# Biceps Brachii and Brachioradialis Excitation in Biceps Curl Exercise: Different Handgrips, Different Synergy

**DOI:** 10.3390/sports11030064

**Published:** 2023-03-09

**Authors:** Giuseppe Coratella, Gianpaolo Tornatore, Stefano Longo, Nicholas Toninelli, Riccardo Padovan, Fabio Esposito, Emiliano Cè

**Affiliations:** 1Department of Biomedical Sciences for Health, Università Degli Studi di Milano, 20133 Milan, Italy; 2IRCCS Galeazzi Orthopedic Institute, 20161 Milan, Italy

**Keywords:** elbow flexors, surface EMG, resistance training, weight training, bodybuilder, strength, root mean square

## Abstract

The current study analyzed the excitation of biceps brachii, brachioradialis, and anterior deltoid during bilateral biceps curl performed with different handgrips. Ten competitive bodybuilders performed bilateral biceps curl in non-exhaustive 6-rep sets using 8-RM with the forearm in supinated, pronated, and neutral positions. The ascending and descending phase of each variation was separately analyzed using the normalized root mean square collected using surface electromyography. During the ascending phase, (i) biceps brachii excitation was greater with the supinated compared to the pronated [+19(7)%, ES: 2.60] and neutral handgrip [+12(9)%, ES: 1.24], (ii) the brachioradialis showed greater excitation with the supinated compared to the pronated [+5(4)%, ES: 1.01] and neutral handgrip [+6(5)%, ES: 1.10], (iii) the anterior deltoid excitation was greater with the pronated and neutral handgrip compared to the supinated condition [+6(3)% and +9(2)%, ES: 2.07 and 3.18, respectively]. During the descending phase, the anterior deltoid showed greater excitation in the pronated compared to the supinated handgrip [+5(4)%, ES: 1.02]. Changing the handgrips when performing biceps curl induces specific variations in biceps brachii and brachioradialis excitation and requires different anterior deltoid interventions for stabilizing the humeral head. Practitioners should consider including different handgrips in the biceps curl routine to vary the neural and mechanical stimuli.

## 1. Introduction

Resistance training is performed to increase strength in consideration of the positive modifications at the neural [1,2] and structural level [3]. All resistance exercises involve muscles primarily carrying out the movement (prime movers) and muscles stabilizing the joint/s around which the movement is realized [4,5,6,7,8,9,10]. Among the resistance exercises targeting the upper limb muscles, the biceps curl is common in practice. The biceps curl is characterized by an elbow flexion accompanied by a dynamic or mainly isometric arm flexion, as well as wrist supination/pronation [5,11]; therefore, the main targeted muscles are the elbow flexors (i.e., biceps brachii, brachioradialis, and brachialis), the arm flexors (e.g., anterior deltoids), and wrist rotators (e.g., biceps brachii, pronator teres). Additionally, biceps curl can be performed with different forms of external resistance like dumbbells, barbells, or iso-load or cable devices [4]. Therefore, the possible combinations of biceps curl variations make this exercise suitable for most resistance training programs since changing one or more factors may provide unique stimuli to the upper limb muscles.

Although not extensive, previous studies have examined the excitation of the main biceps curl prime movers, comparing different variations. For example, when the arm was flexed isometrically at different angles, greater excitation in biceps brachii was observed at longer muscle length [8,12]. As concerns the wrist position, while brachialis has no influence, biceps brachii is a supinator while brachioradialis keeps the forearm in an intermediate position, i.e., between supination and pronation (so-called neutral position). Consequently, also the wrist position (dynamic or isometric) is expected to influence the excitation of the prime movers. The literature in this regard is scanty and the results are controversial. In one previous study, the biceps brachii and brachioradialis were analyzed in the dumbbell curl, straight barbell curl, and EZ-barbell curl (a multi-angle barbell) [10]. The dumbbell curl was performed starting with the hands in a “semi-pronated” position and then ending the ascending phase in supination. The dumbbell curl demonstrated lower activation of both muscles than the EZ-barbell variant in the full exercise [10]. However, the use of different tools during the execution of the exercises and the handgrip changes during the execution of the dumbbell biceps curl introduced two variables hindering the effect of the grip per se on the excitability of biceps brachii and brachioradialis. Comparing straight vs. EZ barbell, the straight induced slightly greater excitation of the biceps brachii, while no information was reported about the brachioradialis [13]. Conversely, no difference in biceps brachii and brachioradial excitation was found while performing dumbbell biceps curl with a supinated and neutral handgrip at different inter-hand distances [14]. Importantly, the pronated forearm handgrip, which is often regarded as the less effective among the biceps curl exercise handgrips, has not received attention so far.

Traditional resistance training includes the performance of both the ascending and the descending phase, often coincident with the concentric and eccentric action of the prime movers. Examining the ascending and descending phase individually, the importance of independently investigating the two phases comes from the distinct acute neuromuscular characteristics [15,16,17], the recovery time course when focusing on the descending phase [18,19,20], and the long-term neuromuscular and structural changes [21,22]. For example, the abovementioned study split the analysis into ascending and descending phases and reported definite distinctions between the exercises [10,13]. In addition, bodybuilders show the ability to perform a given exercise with a consistent technique, increasing the trustfulness of the outcomes [23].

Considering these previous considerations, the aim of the present study was to analyze the level of excitation of the biceps brachii and brachioradialis during the ascending and descending phase of the biceps curl exercise performed with three different handgrips (supinated, pronated, and neutral) at constant hands distance, in a group of competitive bodybuilders. It was hypothesized that the different handgrips would result in different muscle synergies.

## 2. Materials and Methods

### 2.1. Study Design

The present investigation was designed as a cross-over, repeated-measures, within-subject study and was conducted in line with previous studies from our laboratory [13,24,25,26,27,28]. The participants were involved in five different sessions (Figure 1). The sessions were dedicated to the familiarization with the exercise variations (session 1), the determination of the 8-RM for the biceps curl performed using the supinated, pronated, and neutral handgrip (sessions 2–3), and the placement of the electrodes (session 4). In the fifth session, the muscles’ maximum excitation was first measured, and after a 30-min passive recovery, the participants performed a non-exhaustive set for each exercise in a random order, with 10 min of inter-set rest. Each session was separated by at least three days, and the participants were instructed to avoid any further form of resistance training for the entire duration of the investigation.

### 2.2. Participants

We recruited 10 male competitive bodybuilders (age 29.8 ± 3.0 years; body mass 77.9 ± 1.0 kg; stature 1.68 ± 0.01 m; training seniority 10.6 ± 1.8 years) for the present procedures, in line with previous studies [13,24,25,27,28]. They had to be clinically healthy, without any reported history of upper-limb and lower-back muscle injury and neurological or cardiovascular disease in the previous 12 months. To limit possible confounding factors, the participants competed in the same weight category (Men’s Classic Bodybuilding <80 kg, <1.70 m), according to the International Federation of Body Building Pro-League. The participants were required to abstain from alcohol, caffeine, or similar beverages in the 24 h preceding the test. After a full explanation of the aims of the study and the experimental procedures, the participants signed a written informed consent. They were also free to withdraw at any time. The Ethical Committee of the Università degli Studi di Milano (CE 27/17) approved the procedures that were performed following the Declaration of Helsinki (1964 and updates) for studies involving human subjects. The individual in this manuscript has given written informed consent to publish these case details.

### 2.3. Exercises Technique

The biceps curl variations were performed using a Cable Tower (Technogym, Cesena, Italy), with one end of the pulley connected to a bar (42.5 cm length, Technogym, Cesena, Italy) for the supinated and pronated handgrip, and with a rope (Technogym, Cesena, Italy) for the neutral handgrip. The inter-hand distance was measured during the first set and maintained similarly throughout the exercises to limit the effect of the hand’s stance on the muscles’ excitability. Each exercise was performed in a standing position. The arms were maintained parallel to the trunk, with no flexion of the humerus. The curl exercises were characterized by three different handgrips: with the forearm (i) in a supine position, (ii) in a prone position, and (iii) in a neutral position (Figure 2). Whatever the handgrip variation, and following recent updates on the appropriate description of resistance exercise technique [29], the load was fixed as 8-RM, six repetitions were performed—not to failure—to avoid fatigue, with a full range of movement, a timing of 1-2-1-2 s for the first isometric, the ascending, the second isometric and the descending phase respectively (so that all dynamic phases were performed), and using an external focus. The participants were instructed to avoid sagittal oscillations of the trunk, any movement of the lower limbs, and exaggerated elevation of the scapulae, and the technique was checked by three operators. Visual feedback was provided to help the participants follow the timing for each phase [25,26,30]. In case of a lack of consistency in the technique, as described, the participants were required to repeat the set.

### 2.4. 8-RM Procedure

As previously reported, the 8-RM was assessed using the same exercise technique described above [24]. Briefly, after a standardized warm-up consisting of 3 × 15 repetitions of biceps curl exercise using three incremental self-selected loads, the 8-RM was determined incrementing the load until the eighth repetition corresponded to failure, defined as the incapacity to perform the ascending phase [31]. Each attempt was separated by at least 3 min of passive recovery. Operators strongly encouraged the participants to maximally perform each trial.

### 2.5. Maximum Voluntary Isometric Excitation

The area of the appropriate electrode placements was first checked by means of a 13 × 5 semi-disposable high-density electrodes matrix for sEMG detection (GR08MM1305 model, inter-electrode distance of 8 mm, OtBiolettronica, Turin, Italy), as carried out in similar previous studies [13,26]. The sEMG signal was acquired by a multichannel amplifier (EMG-USB model, OtBioelettronica, Turin, Italy; input impedance of >90 MΩ; CMRR of >96 dB; EMG bandwidth of 15–350; gain × 1000). From the analysis of the surface electromyography (sEMG) signal, the innervation zone was identified, and the muscle area involved in the innervation zone shift during the exercises was avoided. Thereafter, the rounded electrodes replaced the high-density electrode matrix.

The maximal voluntary isometric excitation of biceps brachii, brachioradialis, and anterior deltoid was measured in random order following SENIAM procedures [32] using electrodes (mod H124SG Kendall ARBO; diameter: 10 mm; inter-electrodes distance: 20 mm; Kendall, Donau, Germany) equipped with a probe (probe mass: 8.5 g, BTS Inc., Milano, Italy) that permitted the detection and the transfer of the sEMG signal by wireless modality. The sEMG signal was acquired at 1000 Hz, amplified (gain: 2000, impedance and the common rejection mode ratio of the equipment are >1015 Ω//0.2 pF and 60/10 Hz 92 dB, respectively) and driven to a wireless electromyographic system (FREEEMG 300, BTS Inc., Milano, Italy) that digitized (1000 Hz) and filtered (filter type: IV-order Butterworth filter, band-pass 15–350 Hz) the raw sEMG signals. The electrodes were placed on the dominant limb. Each attempt lasted 5 s, and three attempts were completed for each movement interspersed by 3 min of passive recovery [24,28]. For all muscles, the operators provided strong standardized verbal encouragement. In line with previous procedures, the electrodes were placed on the dominant limb [25,28,30].

### 2.6. Data Analysis

The sEMG signals from both the peak value recorded during the maximum voluntary isometric activation and from the ascending and descending phases of each exercise were analyzed in the time domain, using a 25-ms mobile window for the computation of the root mean square (RMS). For the maximum voluntary isometric excitation, the average of the RMS corresponding to the central 2 s was considered. During each exercise, the RMS was calculated and averaged over the 2 s of the ascending and descending phase. To identify the ascending and the descending phase, the sEMG was synchronized with an integrated camera (VixtaCam 30 Hz, BTS Inc., Milano, Italy) that provided the duration of each phase [24,25,30]. Such a duration was used to mark the start and end of each phase while analyzing the sEMG signal. The sEMG data were averaged, excluding the first and the last repetition of each set, to achieve a more consistent technique and decrease the interference of fatigue [33]. Afterward, the sEMG RMS of each muscle during each exercise was normalized (nRMS) for its respective maximum voluntary isometric excitation [24,25,28,30] and inserted into the data analysis.

### 2.7. Statistical Analysis

The statistical analysis was performed using statistical software (SPSS vers. 28.0, IBM, Armonk, NY, USA). The normality of data was checked using the Shapiro–Wilk test and all distributions were normal (*p* > 0.05). Descriptive statistics (participants = 10) are shown as the mean (SD). The differences in the nRMS were separately calculated for the biceps brachii, brachioradialis, and anterior deltoid using a handgrip (3 levels: supination, pronation, and neutral) × phase (2 levels: ascending and descending phase) repeated-measures analysis of variance (ANOVA-RM). Multiple comparisons were adjusted using Bonferroni’s correction and reported as mean difference (SD). Significance was set at α < 0.05. The magnitude of the interactions was calculated using partial eta squared (η_p_^2^) and interpreted as trivial (up to 0.009), small (0.010 to 0.059), medium (0.060 to 0.139), and large (≥0.140) [34]. The pairwise differences are reported with Cohen’s d effect size (ES), which was interpreted according to Hopkins’ recommendations: 0.00–0.19: trivial; 0.20–0.59: small: 0.60–1.19: moderate; 1.20–1.99: large; ≥2.00: very large [35].

## 3. Results

Figure 3 shows the EMG recorded in the biceps brachii. A significant and large handgrip × phase interaction was observed (F_2,18_ = 19.040, *p* < 0.001, η_p_^2^ = 0.679). During the ascending phase, a greater nRMS was observed with the supinated compared to the pronated [+19(7)%, *p* < 0.001, ES: 2.60] and to the neutral handgrip [+12(9)%, *p* = 0.006, ES: 1.24]. Moreover, a greater nRMS was observed with the neutral handgrip compared to the pronated condition [+7(3)%, *p* < 0.001, ES: 2.39]. During the descending phase, no significant differences emerged in nRMS between the three handgrips (*p* > 0.05). nRMS during the ascending phase was greater compared to the descending phase of the corresponding handgrip variation (*p* < 0.001 in all comparisons, ES: 5.84 to 7.73).

Figure 4 shows the EMG recorded in brachioradialis. No handgrip × phase interaction was observed (F_2,18_ = 0.072, *p* = 0.931, η_p_^2^ = 0.008). Significant and large main effects were found for handgrip (F_2_ = 14.954, *p* < 0.001, η_p_^2^ = 0.624) and phase (F_1_ = 35.299, *p* < 0.001, η_p_^2^ = 0.725). During the ascending phase, nRMS was greater with the supinated compared to the pronated [+5(4)%, *p* = 0.007, ES: 1.01] and neutral handgrip [+6(5)%, *p* = 0.004, ES: 1.10]. No significant differences were found between the three handgrips during the descending phase (*p* > 0.05). nRMS during the ascending phase was always greater compared to the descending phase of the corresponding handgrip modality (*p* < 0.001 in all comparisons, ES: 3.03 to 5.35).

Figure 5 shows the EMG recorded in the anterior deltoid. No handgrip × phase interaction was observed (F_2,18_ = 2.601, *p* = 0.102, η_p_^2^ = 0.224), while significant and large main effects were found for handgrip (F_2_ = 26.880, *p* < 0.001, η_p_^2^ = 0.675) and phase (F_1_ = 39.327, *p* < 0.001, η_p_^2^ = 0.774). In the ascending phase, nRMS was greater with the pronated and neutral handgrip compared to the supinated condition [+6(3)% and +9(2)%, *p* < 0.001 for both comparisons, ES: 2.07 and 3.18, respectively]. Interestingly, during the descending phase, a greater nRMS was found in the pronated condition compared to the supinated handgrip [+5(4)%, *p* = 0.020, ES: 1.02]. Likewise, in the anterior deltoid, regardless of the handgrip variation, the nRMS in the ascending phase was greater compared to the descending phase (*p* < 0.001 in all comparisons, ES: 2.38 to 3.34).

## 4. Discussion

The aim of the current study was to analyze, in a group of competitive bodybuilders, the level of excitation of the biceps brachii and brachioradialis during the ascending and descending phase of the biceps curl performed with three different handgrips (supinated, pronated, and neutral) keeping constant the inter-hand distance. The results show that both biceps brachii and brachioradialis exhibited greater levels of excitation with the supinated compared to the pronated and neutral handgrip. Moreover, the lowest biceps brachii excitation was found with the forearm pronated, while the excitation of brachioradialis was similar in the neutral and pronated condition. These differences appeared only during the ascending phase, while no difference was visible during the descending phase. Interestingly, the anterior deltoid exhibited greater excitation with a pronated and neutral than the supinated handgrip. Regardless of the handgrip, the level of muscle excitability was greater in the three muscles during the ascending than in the descending phase. The supinated seems to excite the elbow flexors more than the neutral and pronated handgrip while requiring less stabilization of the glenohumeral joint.

Before exploring the literature, deepening some anatomical characteristics of the muscles examined here may provide some support in explaining our results. The biceps brachii is the largest anterior muscle of the arm and originates with two heads from the scapula and inserts with a common tendon to the tuberosity of the radius. While performing the biceps curl: (i) flexes the elbow and supinate or stabilizes the forearm towards the supination, and (ii) flexes or stabilizes the humeral head anteriorly. Brachioradialis originates from the humeral supracondylar crest and lateral epicondyle and inserts into the styloid process of the radius. It flexes the elbow and keeps the forearm in a neutral position. Although not examined here because only detectable through wire electrodes, brachialis also has a key role in elbow flexion. Brachialis is the most powerful flexor of the forearm and, unlike the biceps brachii and brachioradialis, does not insert on the radius, therefore not participating in the forearm supination.

The novel aspect of this study is the inclusion of the pronated handgrip to compare the excitation of the elbow flexors while performing the biceps curl, while previous studies have limited the comparison to the supinated vs. neutral or semi-supinated handgrip. As concerns the biceps brachii, the excitation was supinated > neutral > pronated throughout the ascending phase. Although a straight comparison with previous studies cannot be made, we recently observed that both dynamic and stable supination increased the biceps brachii excitation while comparing a series of upper limb exercises [6], and interestingly dynamic supination appeared even more effective than a forearm in a stable supinated position [36] In line, a slightly greater excitation of the biceps brachii was seen in the supinated handgrip using the straight barbell compared a semi-supinated handgrip using the EZ barbell [13]. In contrast, other authors did not find any difference between the two, albeit the different load and population may have affected the results [10,14]. Overall, the biceps brachii seem more excited with the increase in forearm supination. This could also suggest a higher degree of tension of the biceps brachii long head tendon, which would confer greater stabilization at the level of the humerus head. Notwithstanding, the SENIAM recommendations for the location of the electrodes on the biceps brachii do not permit separating the short and the long head [32,37], so the results refer to the biceps brachii as a whole.

Surprisingly, brachioradialis showed greater excitability with the supinated grip than with the neutral and pronated grip, with no difference between these two. Such outcomes may be in contrast with the general idea that biceps curl with a neutral handgrip should increase the role of brachioradialis. This anecdotal belief has been questioned in previous studies, where no difference was observed comparing the straight vs. the EZ barbell [10,14]. However, it should be noticed that the handgrip position of the EZ barbell is rather semi-supinated than neutral, maybe not enough to elicit any difference. In the first instance, as previously shown when comparing different variations of the same exercise, a muscle in a more lengthened position results in a greater amplitude of the sEMG signal [8,12,24,28,38,39]. As in the present case, brachioradialis is more lengthened using the supinated than the neutral grip. Nevertheless, one may expect that this should also occur for the pronated grip, where the brachioradialis is lengthened as well. However, the pronated grip is expected to require greater wrist stabilization, especially towards the wrist extension, so as to involve further muscles such as the extensor carpi and other wrist extensors. Such a synergy may have advantaged the lifting of the external load, possibly making unnecessary an over-excitation of the brachioradialis.

Even though the biceps curl was performed without a dynamic arm flexion, the anterior deltoid must stabilize the glenohumeral joint during the whole movement. Both the pronated and neutral showed greater excitation compared to the supinated handgrip. When considering the whole between-muscle synergy, this might have been compensation for the lower excitation of the biceps brachii, especially its long head, to stabilize the humeral head. This behavior is in apparent contradiction with recent results that found greater excitation of the anterior deltoid with the straight compared to the EZ-barbell [13]. However, to better accommodate the anatomical valgus of the elbow with the EZ-barbell, the inter-hand distance must be shorter than with the straight barbell. This may have led to a greater external rotation of the humerus with a greater lengthening of the anterior deltoid fibers, thus possibly promoting an increase in its excitation [13,38,39].

The descending phase showed overall lower excitation for all muscles compared to the ascending phase, as previously observed in this kind of study [13,24,25,26,27,28]. The descending phase corresponds to the eccentric contraction for all the muscles examined, and it is well known that active lengthening makes a greater force exertion and, consequently, a greater external load [21,40,41]. This may depend on both neural [15,16,17] and structural factors [42,43,44] that favor eccentric rather than concentric force exertion. Therefore, a given load is less demanding during the descending than ascending phase. Interestingly the between-variation differences in biceps brachii and brachioradialis excitation observed during the ascending disappeared during the descending phase, while the anterior deltoid was still more excited during the pronated handgrip. As a whole, it is possible the motor control used for the descending phase shifted toward greater stabilization of the humeral head so to explain the behavior of the anterior deltoid. This appears to consent to a similar biceps brachii and brachioradialis intervention once the humerus is stabilized.

The present investigation comes with some acknowledged limitations. First, the excitation of the flexor and extensor carpi was not recorded and would have helped to interpret the load stabilization during the three biceps curl variations [45]. Second, the pectoralis major was not examined, so more information about the humerus head stabilization is needed. Third, the present results refer to the present population and sample size and the detailed technique we used; hence they should not be generalized. Last, the biceps curl encompasses further variations to be investigated.

## 5. Conclusions

The supinated handgrip elicited the greatest excitation for both biceps brachii and brachioradialis during the ascending phase, while no between-handgrip difference was found during the descending phase. The anterior deltoid was more excited during the pronated and neutral than the supinated handgrip, and the pronated was still greater than the supinated handgrip during the descending phase. Lastly, regardless of the handgrip, the ascending excited all three muscles more than the descending phase.

When providing some practical application, the diversification of the stimuli using different handgrips should be considered when aiming to reinforce the upper limb muscles. This may imply a combination of both neuromuscular and mechanical stress that may assist the overload progression or just the variation of the training session. Interestingly, one may also perform the ascending or descending phase together, as in traditional resistance training, or separately to take advantage of the distinct neural and mechanical patterns.

## Figures and Tables

**Figure 1 sports-11-00064-f001:**
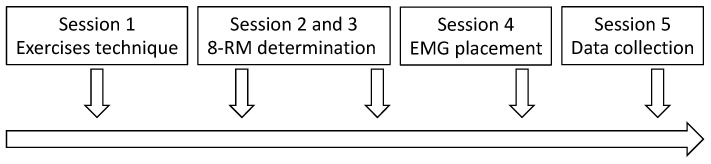
Schematic representation of the study design.

**Figure 2 sports-11-00064-f002:**
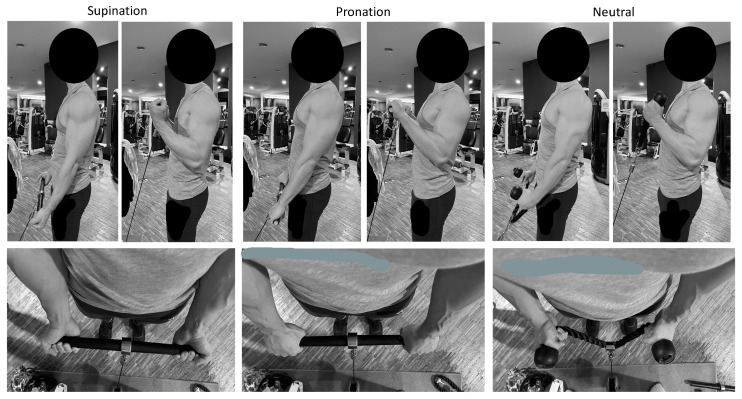
Initial and final position and handgrip variation during the three biceps curl variation: supinated (**panels on the left**), pronated (**middle panels**), and neutral handgrip (**panels on the right**).

**Figure 3 sports-11-00064-f003:**
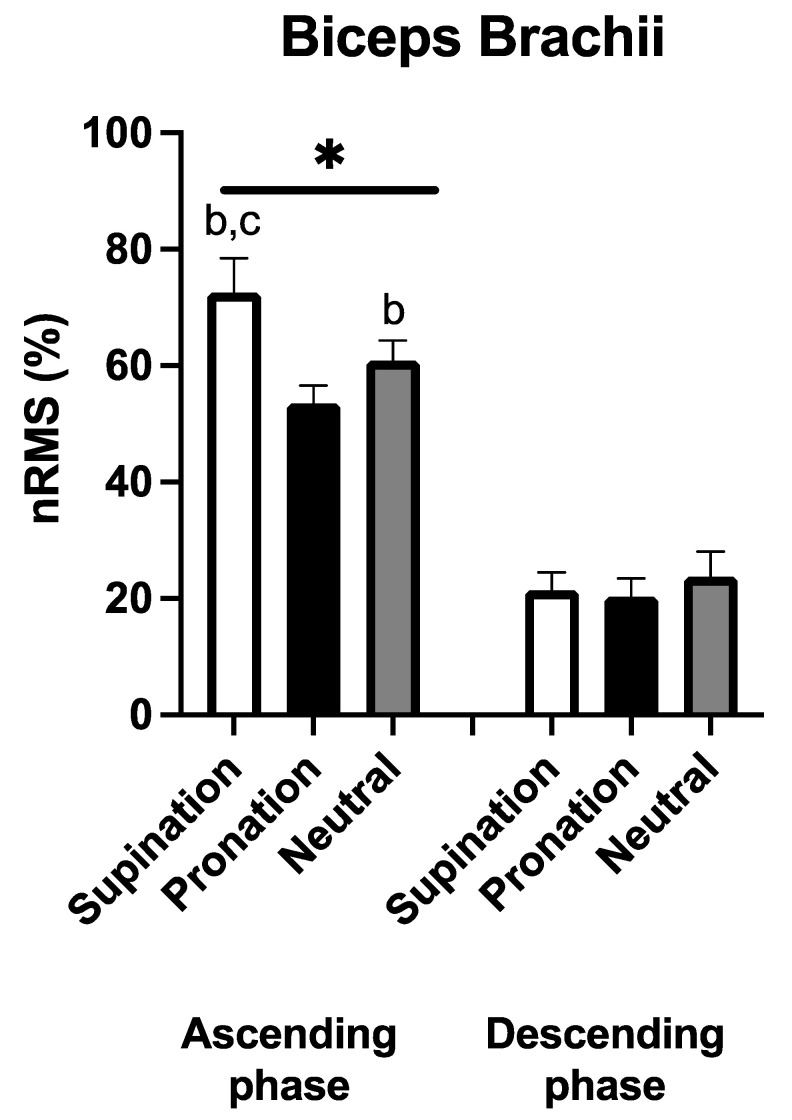
Average (N = 10) normalized root mean square (nRMS) for the biceps brachii during the three different handgrips in the ascending and descending phase. b: *p* < 0.05 vs. pronation, c: *p* < 0.05 vs. neutral, *: *p* < 0.05 vs. the descending phase.

**Figure 4 sports-11-00064-f004:**
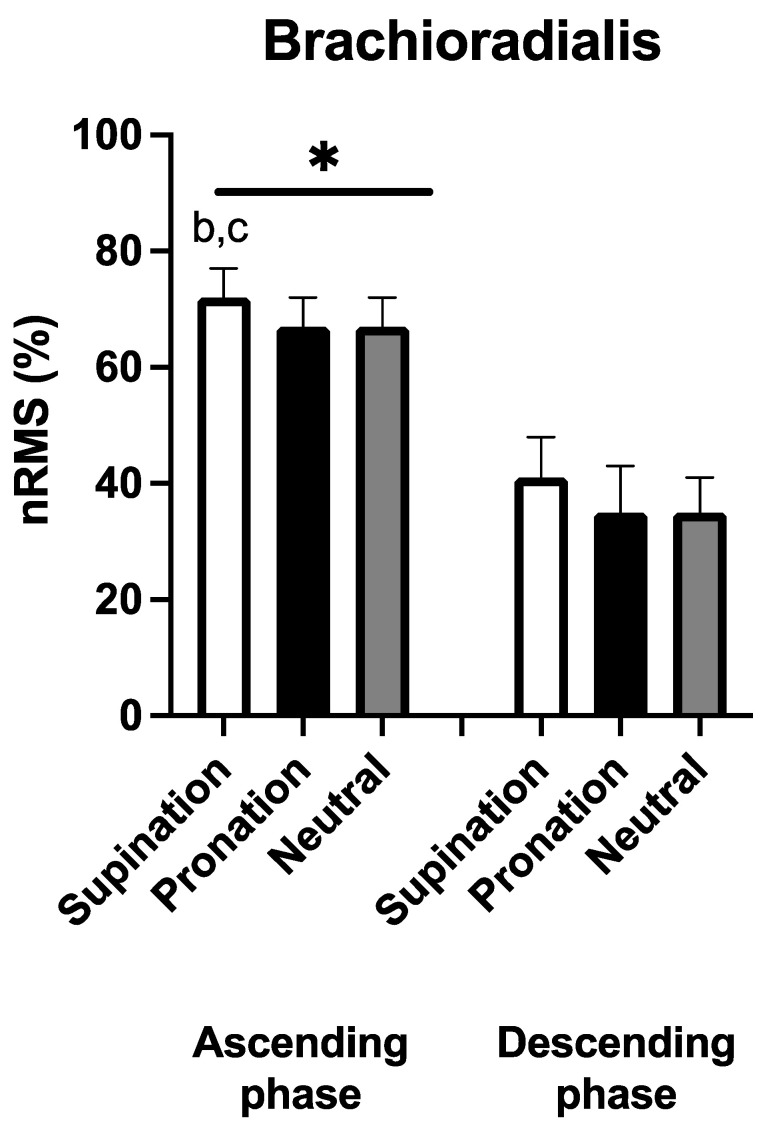
Average (N = 10) normalized root mean square (nRMS) for the brachioradialis during the three different handgrips in the ascending and descending phase. b: *p* < 0.05 vs. pronation, c: *p* < 0.05 vs. neutral, *: *p* < 0.05 vs. the descending phase.

**Figure 5 sports-11-00064-f005:**
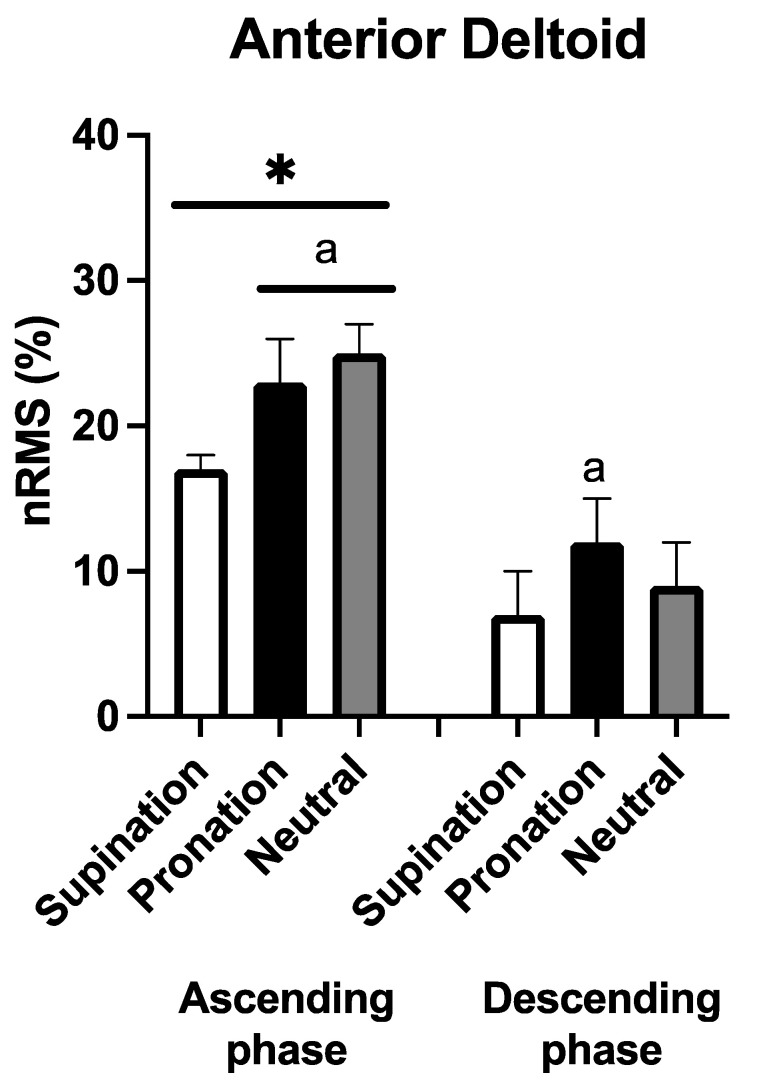
Average (N = 10) normalized root mean square (nRMS) for anterior deltoid during the three different handgrips in the ascending and descending phase. a: *p* < 0.05 vs. supination; *: *p* < 0.05 vs. the descending phase.

## Data Availability

Data are available on request to the corresponding author.

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
