# Peer review of "Biceps Brachii and Brachioradialis Excitation in Biceps Curl Exercise: Different Handgrips, Different Synergy"

_sports, 2023, doi:10.3390/sports11030064_

Round 1

Reviewer 1 Report

Dear Authors

As one of the reviewers, I express my personal scientific opinion on your work. I would like to reassure you that I was trying to be positive and constructive but particularly as fair and honest to your work. The clear explanation provided in Methods section including the Study design schematic representation is appreciated. I should also note that the originality of the study, the statistical approach, using Bonferroni adjustment particularly, the calculation of the magnitude of interactions and of the Effect Size as well as the excellent work done on figures and the presentation of the limitations’ section, are all positive points. Yet, the lack of test-retest reliability and/or intra-class correlation coefficient (ICC) and confident intervals (CI) are somewhat negative points.

Please accept my judgment with a positive and constructive way.

1.      Line 53: What is the EZ-barbell? You need perhaps to explain (…it is a multi-angled specific barbell…)?

2.      What is actually the Hypothesis of the study?

3.      Lines 83 and 116, and throughout the text: In my point of view and particularly for avoiding self-citations, you should consider to avoid mentioning all your previous published studies in the current one.

4.      Limitations: It is strongly appreciated that you had mentioned this in your limitation section. However, why you have not evaluated, 1) the excitation of the flexor and extensor carpi muscles and 2) the excitation of the pectoralis major?

Author Response

Reviewer 1

As one of the reviewers, I express my personal scientific opinion on your work. I would like to reassure you that I was trying to be positive and constructive but particularly as fair and honest to your work. The clear explanation provided in Methods section including the Study design schematic representation is appreciated. I should also note that the originality of the study, the statistical approach, using Bonferroni adjustment particularly, the calculation of the magnitude of interactions and of the Effect Size as well as the excellent work done on figures and the presentation of the limitations’ section, are all positive points. 

Response: We thank the reviewer

Yet, the lack of test-retest reliability and/or intra-class correlation coefficient (ICC) and confident intervals (CI) are somewhat negative points.

Response: We agree with the reviewer on the appropriateness of calculating the intraclass correlation coefficient. However, it was not conducted in this study given that we found high repeatability in two of our previous studies involving the same subjects and using a setup similar to the one used here (Coratella et al., Eur J Sport Sci, 2020 Jun;20(5):571-579 and Coratella et al., Int J Environment Res Public Health, 2020;17:6015).  Therefore, we are reasonably confident in the solidity and robustness of the approach used.

Please accept my judgment with a positive and constructive way.

  1. Line 53: What is the EZ-barbell? You need perhaps to explain (…it is a multi-angled specific barbell…)?

Response: We have described the EZ-barbell as requested.

  1. What is actually the Hypothesis of the study?

Response: We have added the hypothesis.

  1. Lines 83 and 116, and throughout the text: In my point of view and particularly for avoiding self-citations, you should consider to avoid mentioning all your previous published studies in the current one.

Response: In many points these citations are methodological, just to ensure the scientific validity of the procedures used here.

  1. Limitations: It is strongly appreciated that you had mentioned this in your limitation section. However, why you have not evaluated, 1) the excitation of the flexor and extensor carpi muscles and 2) the excitation of the pectoralis major?

Response: We agree with this reviewer. Unfortunately, we had only limited materials at the time of the data collection, and we chose the muscles in the study, excluding other muscles.

Reviewer 2 Report

Thank you for the opportunity to review the manuscript. The study has practical relevance for sports and is therefore suitable for sports. One comment on the paper:

The abstract should be set in justified text. The hyphenation should be checked. The introduction is well formulated and the central question is addressed. The methodology of the study is well written and comprehensible. The description of the conduct of the study, the execution of the exercise, the sample of subjects, the EMG analysis, the statistical evaluation is also well implemented. Figure 3 and legend should be on one page. The summary should also be set in block capitals.

In terms of content, it should be pointed out that in terms of varied training stimuli, many different exercise designs can be used depending on the objective. Thus, all variations would have their justification. 

Author Response

Reviewer 2

Thank you for the opportunity to review the manuscript. The study has practical relevance for sports and is therefore suitable for sports.

Response: We thank the reviewer.

One comment on the paper:

The abstract should be set in justified text.

Response: Done.

The hyphenation should be checked.

Response: Done.

The introduction is well formulated and the central question is addressed.

Response: We thank the reviewer.

The methodology of the study is well written and comprehensible.

Response: We thank the reviewer.

The description of the conduct of the study, the execution of the exercise, the sample of subjects, the EMG analysis, the statistical evaluation is also well implemented.

Response: We thank the reviewer.

Figure 3 and legend should be on one page. The summary should also be set in block capitals.

Response: We believe that, in case of acceptance, this will be edited by the Editorial Office.

In terms of content, it should be pointed out that in terms of varied training stimuli, many different exercise designs can be used depending on the objective. Thus, all variations would have their justification. 

Response: Reported in the practical applications.

Reviewer 3 Report

The authors present a good research paper. 

  • The relevance of the topic: Good.
  • Introduction: Can be improved.
  • Methodology: Can be improved.
  • Results: Good.
  • Discussion: Good.  

However, ACCEPT AFTER MINOR REVISION. In general, the paper follows an adequate structure and correct scientific support and can be published considering some limitations. The study is interesting in the field of muscle exercises. However, there are a series of limitations that should be considered.

In the first place, carry out a review of the existing literature related to the subject, being essential to inquire into the MPDI – Sports journal itself, since there are papers related to its manuscript that can help to improve it. Therefore, include those references, if any, especially from the last five years. In addition, recommend reading some papers related to the topic of muscle exercises:

Nunes, J. P., Jacinto, J. L., Ribeiro, A. S., Mayhew, J. L., Nakamura, M., Capel, D. M., ... & Aguiar, A. F. (2020). Placing greater torque at shorter or longer muscle lengths? Effects of cable vs. barbell preacher curl training on muscular strength and hypertrophy in young adults. International Journal of Environmental Research and Public Health17(16), 5859.

Specific comments.

Title. The title of the manuscript is correct.

Abstract. Incorporate in the summary, a more precise sentence of the results.

Introduction. This section presents the problem in a coherent and clear manner with the correct support of the scientific literature. However, it is convenient to update the references, since there are different documents related to the subject and no mention is made, and it would even be interesting to mention the different existing studies related to muscle exercises. Also, it could be a future study of review. Some bibliographical references are attached to carry out the section of muscle exercises:

Ha, S. Y., & Shin, D. (2020). The effects of curl-up exercise in terms of posture and muscle contraction direction on muscle activity and thickness of trunk muscles. Journal of Back and Musculoskeletal Rehabilitation33(5), 857-863.

Methods. Modify the method section, and specifically, in the section: Design.

-       Study design. To write the design section, we recommend that you take some of the following methodologists as references.

Ato, M., López-García, J. J., & Benavente, A. (2013). A classification system for research designs in psychology. Annals of Psychology29(3), 1038-1059.

Results. Summary of study data and table are correct.

Discussion. The section Discussion is correct.

Conclusion. Differentiate the discussion of the main conclusions of the study. To do this, you must create this section. And modify the limitations of the study and locate them in said section at the end. Also, they must be direct, and highlight the main contributions of the study.

References. They should be reviewed and updated according to the publication standards. There are many errors in the references. Therefore, correct them and adapt them to the magazine's regulations.

Author Response

Reviewer 3

The authors present a good research paper. 

 The relevance of the topic: Good.

Introduction: Can be improved.

Methodology: Can be improved.

Results: Good.

Discussion: Good.  

 However, ACCEPT AFTER MINOR REVISION. In general, the paper follows an adequate structure and correct scientific support and can be published considering some limitations. The study is interesting in the field of muscle exercises. However, there are a series of limitations that should be considered.

Response: We thank the reviewer for the suggestions. We hope the current version might reach the desired quality.

In the first place, carry out a review of the existing literature related to the subject, being essential to inquire into the MPDI – Sports journal itself, since there are papers related to its manuscript that can help to improve it. Therefore, include those references, if any, especially from the last five years. In addition, recommend reading some papers related to the topic of muscle exercises: Nunes, J. P., Jacinto, J. L., Ribeiro, A. S., Mayhew, J. L., Nakamura, M., Capel, D. M., ... & Aguiar, A. F. (2020). Placing greater torque at shorter or longer muscle lengths? Effects of cable vs. barbell preacher curl training on muscular strength and hypertrophy in young adults. International Journal of Environmental Research and Public Health17(16), 5859.

Response: Actually, we the reference the reviewer recommended was already in the reference list. As suggested, we have added a relevant paper from the Journal.

Response: Specific comments.

Title. The title of the manuscript is correct.

Response: We thank the reviewer.

Abstract. Incorporate in the summary, a more precise sentence of the results.

Response: Actually not clear what “precise” means here, but please consider the wordcount limit for the abstract.

Introduction. This section presents the problem in a coherent and clear manner with the correct support of the scientific literature. However, it is convenient to update the references, since there are different documents related to the subject and no mention is made, and it would even be interesting to mention the different existing studies related to muscle exercises. Also, it could be a future study of review. Some bibliographical references are attached to carry out the section of muscle exercises: Ha, S. Y., & Shin, D. (2020). The effects of curl-up exercise in terms of posture and muscle contraction direction on muscle activity and thickness of trunk muscles. Journal of Back and Musculoskeletal Rehabilitation33(5), 857-863.

Response: We have updated the reference list as suggested.

Methods. 

Modify the method section, and specifically, in the section: Design.  Study design. To write the design section, we recommend that you take some of the following methodologists as references: Ato, M., López-García, J. J., & Benavente, A. (2013). A classification system for research designs in psychology. Annals of Psychology29(3), 1038-1059.

Response: We believe that there is no need of this specific reference in this context.

Results. Summary of study data and table are correct.

Response: We thank the reviewer.

Discussion. The section Discussion is correct.

Response: We thank the reviewer.

Conclusion. Differentiate the discussion of the main conclusions of the study. To do this, you must create this section. And modify the limitations of the study and locate them in said section at the end. Also, they must be direct, and highlight the main contributions of the study.

Response: We respectfully disagree with the reviewer about the limitations location, which we believe should stay where they are. We have cut some non-useful sentences to be more straightforward.

References. They should be reviewed and updated according to the publication standards. There are many errors in the references. Therefore, correct them and adapt them to the magazine's regulations.

Response: We have double-checked the bibliography style, considering that it is automatically provided by a dedicated software.